# ANALYZING POPULATIONS OF NEURAL NETWORKS VIA DYNAMICAL MODEL EMBEDDING

## ABSTRACT

A core challenge in the interpretation of deep neural networks is identifying commonalities between the underlying algorithms implemented by distinct networks trained for the same task. Motivated by this problem, we introduce DYNAMO, an algorithm that constructs low-dimensional manifolds where each point corresponds to a neural network model, and two points are nearby if the corresponding neural networks enact similar high-level computational processes. DYNAMO takes as input a collection of pre-trained neural networks and outputs a *meta-model* that emulates the dynamics of the hidden states as well as the outputs of any model in the collection. The specific model to be emulated is determined by a *model embedding vector* that the meta-model takes as input; these model embedding vectors constitute a manifold corresponding to the given population of models. We apply DYNAMO to both RNNs and CNNs, and find that the resulting model embedding spaces enable novel applications: clustering of neural networks on the basis of their high-level computational processes in a manner that is less sensitive to reparameterization; model averaging of several neural networks trained on the same task to arrive at a new, operable neural network with similar task performance; and semi-supervised learning via optimization on the model embedding space. Using a fixed-point analysis of meta-models trained on populations of RNNs, we gain new insights into how similarities of the topology of RNN dynamics correspond to similarities of their high-level computational processes.

## 1 INTRODUCTION

A crucial feature of neural networks with fixed network architecture is that they form a manifold by virtue of their continuously tunable weights, which underlies their capability to be trained by gradient descent. However, this conception of the space of neural networks is inadequate for understanding the computational processes they perform. For example, two neural networks trained to perform the same task may have vastly different weights, and yet implement the same high-level algorithms and computational processes (Maheswaranathan et al., 2019b).

In this paper, we construct an algorithm which provides alternative parametrizations of the space of RNNs and CNNs with the goal of endowing a geometric structure that is more compatible with the high-level computational processes performed by neural networks. In particular, given a set of neural networks with possibly different architectures (and possibly trained on different tasks), we find a parametrization of a low-dimensional submanifold of neural networks which approximately interpolates between these chosen "base models", as well as extrapolates beyond them. We can use such *model embedding spaces* to cluster neural networks and even compute *model averages* of neural networks. A key feature is that two points in model embedding space are nearby if they correspond to neural networks which implement similar high-level computational processes, in a manner to be described later. In this way, two neural networks may correspond to nearby points in model embedding space even if those neural networks have distinct weights or even architectures.

The model embedding space is parametrized by a low-dimensional parameter $\theta \in \mathbb{R}^d$, and each base model is assigned a value of $\theta$ in the space. This allows us to apply traditional ideas from clustering and interpolation to the space of neural networks. Moreover, each model embedding space has an associated *meta-model* which, upon being given a $\theta$, is rendered into an operable neural network. If a base model is mapped to some $\theta$, then the meta-model, upon being given that $\theta$, will emulate the

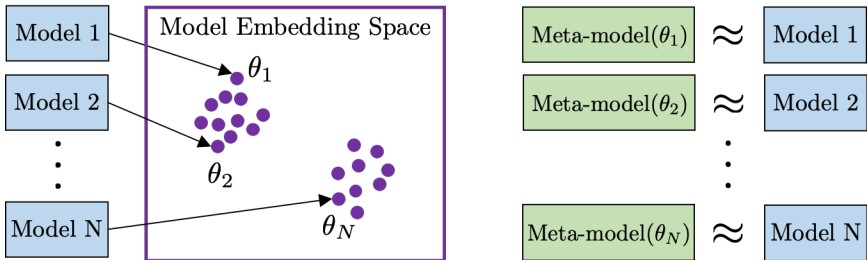

Figure 1: A conceptual diagram of the DYNAMO algorithm and *model embedding space*. A set of neural networks labelled $1, ..., N$ called the *base models* are mapped to corresponding points $\theta_1, ..., \theta_N$ in the model embedding space. Two points in the model embedding space are nearby if they correspond to neural networks with similar dynamics for their hidden and output states. There is an associated neural network called the *meta-model* which, when given a $\theta_i$, becomes a neural network that emulates the hidden and output dynamics of the corresponding base model. The meta-model produces a viable neural network for any given any value of $\theta$, including points in the model embedding space that do not correspond to any base model.

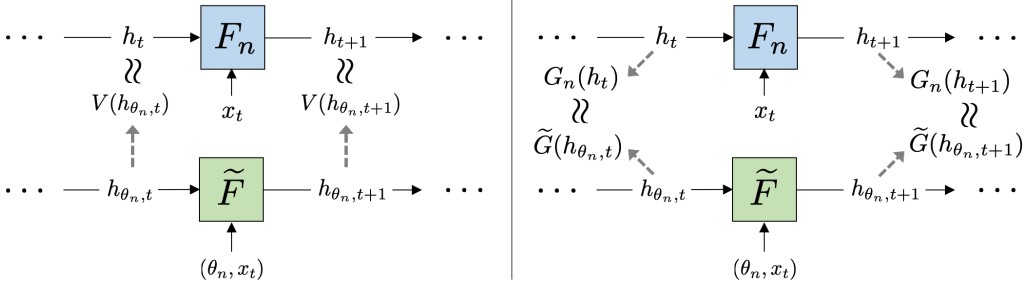

Figure 2: **Left:** The hidden states $h_t$ of $F_n$ are close to the hidden states $h_{\theta_n,t}$ of $\widetilde{F}$ after the transformation map $V$ is applied. **Right:** The visible states $G(h_t)$ of $F_n$ are close to the visible states $\widetilde{G}(h_{\theta_n,t})$.

corresponding base model. See Figure 1 for a diagrammatic depiction. An interesting application is that given two models assigned to parameters $\theta_1$ and $\theta_2$, we can consider the averaged model corresponding to the value $(\theta_1 + \theta_2)/2$. We find that this averaged model performs similarly on the task for which the base models in the average were trained. We also use the model embedding space to extrapolate outside the space of base models, and find cases in which the model embedding manifold specifies models that perform better on a task than any trained base model. Later on we will explain how this can be regarded as a form of semi-supervised learning.

The rest of the paper is organized as follows. We first provide the mathematical setup for our algorithmic construction of model embedding spaces. After reviewing related work, we then present results of numerical experiments which implement the algorithm and explore clustering, model averaging, and semi-supervised learning on the model embedding space. We further examine how topological features of the dynamics of RNNs in model embedding spaces are reflective of classes of high-level computation processes. Finally, we conclude with a discussion.

## 2 DYNAMICAL MODEL EMBEDDING

### 2.1 MATHEMATICAL SETUP

In this section, we provide the mathematical setup for construction of model embedding spaces. We treat this in the RNN setting, and relegate the CNN setting to Section A of the Appendix.

---

**Algorithm 1** DYNAMO algorithm

---

**Input:** Base models $\{(F_n, G_n)\}_{n=1}^N$, loss functions $\{(\mathcal{L}_{n,\text{hidden}}, \mathcal{L}_{n,\text{output}})\}_{n=1}^N$,
    datasets $\{\mathcal{D}_n\}_{n=1}^N$, output loss weight $\lambda \geq 0$
**Output:** Meta-model $(\widetilde{F}, \widetilde{G})$, state maps $\{V_n\}_{n=1}^N$, model embeddings $\{\theta_n\}_{n=1}^N$

Initialize networks $\widetilde{F}, \widetilde{G}, \{V_n\}_{n=1}^N$
Initialize model embeddings $\theta_n = 0$
**for** $\tau = 1, \ldots, \tau_{\max}$ **do**
    Sample base model $(F_i, G_i)$ uniformly at random
    Sample input $\{x_t\}$ from dataset $\mathcal{D}_i$
    Compute base model hidden states $\{h_t\}$ and outputs $\{G_i(h_t)\}$
    Compute meta-model hidden states $\{h_{\theta_i,t}\}$ and outputs $\{\widetilde{G}(h_{\theta_i,t})\}$
    Compute mapped meta-model states $\{V_i(h_{\theta_i,t})\}$
    Compute loss $\widehat{\mathcal{L}} = \mathcal{L}_{i,\text{hidden}} + \lambda\,\mathcal{L}_{i,\text{output}}$
    Update $\widetilde{F}, \widetilde{G}, V_i, \theta_i$ using $\nabla\widehat{\mathcal{L}}$
**end for**
**return** $(\widetilde{F}, \widetilde{G}), \{V_n\}, \{\theta_n\}$

---

**Notation.** Let an RNN be denoted by $F(x, h)$ where $x$ is the input and $h$ is the hidden state. We further denote the hidden-to-output map by $G(h)$. We consider a collection of $N$ RNNs $\{(F_n, G_n)\}_{n=1}^N$ we call the *base models* which may each have distinct dimensions for their hidden states, but all have the same dimension for their inputs as well as the same dimension for their outputs. A sequence of inputs is notated as $\{x_t\}_{t=0}^T$ which induces a sequence of hidden states by $h_{t+1} = F(x_t, h_t)$ where the initial hidden state $h_0$ is given. A collection of sequences of inputs, possibly each with different maximum lengths $T$, is denoted by $\mathcal{D}$ which we call an *input data set*. We suppose that each base model RNN has an associated input data set.

## 2.2 META-MODELS FOR RNNS

Given a collection base model RNNs, we would like to construct a *meta-model* which emulates the behavior of each of the base models. In this case, the meta-model is itself an RNN with one additional input $\theta \in \mathbb{R}^d$ and a corresponding map $\widetilde{F}(\theta, x, h)$ whose output is the next hidden state. Given a sequence of input states $\{x_t\}_{t=0}^T$, we have a corresponding sequence of output states $h_{\theta,t+1} = \widetilde{F}(\theta, x_t, h_{\theta,t})$ starting from an initial hidden state $h_0$ (which we suppose does not depend on $\theta$). The meta-model also includes a hidden-to-output map $\widetilde{G}(h)$ that is independent of $\theta$.

For the meta-model $(\widetilde{F}, \widetilde{G})$ to emulate a particular base model $(F_n, G_n)$ with respect to its corresponding data set $\mathcal{D}_n$, we consider the following criteria: there is some $\theta_n$ for which

1. $\widetilde{G}(h_{\theta_n,t}) \approx G_n(h_t)$ for all $t > 0$ and all input sequences in the data set

2. $V_n(h_{\theta,t}) \approx h_t$ for all $t > 0$ and all input sequences in the data set,

where $V_n$ is a transformation of a meta-model's hidden activity. We emphasize that $\theta_n$ and $V_n$ depend on the particular base model under consideration. The first criterion means that at some particular $\theta_n$, the outputs of the meta-model RNN dynamics are close to the outputs of the base model RNN dynamics. The second criterion means that at the same $\theta_n$, there is a time-independent transformation $V_n$ (i.e., $V_n$ does not depend on $t$) such the transformed hidden state dynamics of the meta-model are close to the hidden state dynamics of the base model. See Figure 2 for a visualization. As depicted in the figure, it is convenient to regard the meta-model RNN as having input states $(\theta_n, x)$. As such, a sequence of inputs $\{x_t\}_{t=1}^T$ is appended by $\theta_n$ to become $\{(\theta_n, x_t)\}_{t=1}^T$.

The desired properties of the meta-model are enforced in the loss function. Defining the functions

$$\mathcal{L}_{\text{output}}[\widetilde{F}, \widetilde{G}, \theta_n] := \frac{1}{T} \sum_{t=1}^{T} d(\widetilde{G}(h_{\theta_n, t}), G_n(h_t)) \tag{1}$$

$$\mathcal{L}_{\text{hidden}}[\widetilde{F}, \theta_n, V_n] := \frac{1}{T} \sum_{t=1}^{T} \|V_n(h_{\theta_n, t}) - h_t\|_2^2 \tag{2}$$

where $d$ is some suitable distance or divergence, we can construct the loss function

$$\mathbb{E}_{\{x_t\} \sim \mathcal{D}_n} \left[ \mathcal{L}_{\text{hidden}}[\widetilde{F}, \theta_n, V_n] + \lambda \, \mathcal{L}_{\text{output}}[\widetilde{F}, \widetilde{G}, \theta_n] \right] \tag{3}$$

where we average over the choice of sequence $\{x_t\}$ coming from the input data set $\mathcal{D}_n$. Above, $\lambda$ is a hyperparameter. Our aim is to minimize the above over a suitable class of $\widetilde{F}, \widetilde{G}, V_n$, as well as $\theta_n$; this can be implemented computationally via the DYNAMO algorithm (see Algorithm 1). As a side remark, it appears that a suitable alternative choice to equation 2 would be $\frac{1}{T} \sum_{t=1}^{T} \|h_{\theta_n, t} - W_n(h_t)\|_2^2$ where here $W_n$ is a map from the hidden states of the base model to the hidden states of the meta-model. However, this would be problematic since minimization may pressure $W_n$ to be the the zero map (or otherwise have outputs which are small in norm) and accordingly pressure the dynamics of the meta-model to be the trivial (or have small norm). As such, we opt to formulate $\mathcal{L}_{\text{hidden}}$ as it is written in equation 2.

Suppose we want the meta-model to be able to emulate an entire collection of base models $\{(F_n, G_n)\}_{n=1}^{N}$. In particular, the meta-model will attempt to assign to the $n$th base model a $\theta_n$ and $V_n$ so that the two criteria listed above are satisfies for that base model. These desiderata can be implemented by minimizing the loss function

$$\mathcal{L}[\widetilde{F}, \widetilde{G}, \{\theta_n\}_{n=1}^{N}, \{V_n\}_{n=1}^{N}] := \frac{1}{N} \sum_{n=1}^{N} \mathbb{E}_{\{x_t\} \sim \mathcal{D}_n} \left[ \mathcal{L}_{n,\text{hidden}}[\widetilde{F}, \theta_n, V_n] + \lambda \, \mathcal{L}_{n,\text{output}}[\widetilde{F}, \widetilde{G}, \theta_n] \right]. \tag{4}$$

In some circumstances, we may want to consider base models with distinct dimensions for their output states. For instance, suppose half of the base models perform a task with outputs in $\mathbb{R}^{d_1}$ and the other half perform a task without outputs in $\mathbb{R}^{d_2}$. To accommodate for this, we can have two hidden-to-output maps $\widetilde{G}_1, \widetilde{G}_2$ for the meta-model, where the maps have outputs in $\mathbb{R}^{d_1}$ and $\mathbb{R}^{d_2}$ respectively. The loss function is slightly modified so that we use $\widetilde{G}_1$ where compare the meta-model to the first kind of base model, and $\widetilde{G}_2$ when we compare the meta-model to the second kind of base model. This construction generalizes to the setting where the base models can be divided up into $k$ groups with distinct output dimensions; this would necessitate $k$ hidden-to-output functions $\widetilde{G}_1, ..., \widetilde{G}_k$ for the meta-model.

## 3   RELATED WORK

There is a substantial body of work on interpreting the computational processes implemented neural networks by studying their intermediate representations. Such analyses can be performed either on individual models (Simonyan et al., 2013; Zeiler & Fergus, 2014; Lenc & Vedaldi, 2015) or on collections of models (Li et al., 2016; Raghu et al., 2017; Morcos et al., 2018; Kornblith et al., 2019). Prior work in this latter category has focused on pairwise comparisons between models. For example, SVCCA (Raghu et al., 2017) uses canonical correlation analysis to measure the representational similarity between pairs of models. While these methods can also be used to derive model representations by embedding the pairwise distance matrix, our approach does not require the $\Theta(N^2)$ computational cost of comparing all pairs of base models. Moreover, DYNAMO yields an executable meta-model that can be called with model embedding vectors other than those corresponding to the base models.

There is a related body of work in the field of computational neuroscience. CCA based techniques and representational geometry are standard for comparing neural networks to the neural activations of animals performing vision tasks (Yamins & DiCarlo, 2016) as well as motor tasks. In an example

of the latter, the authors of (Sussillo et al., 2015) used CCA techniques to compare brain recordings to those of neural networks trained to reproduce the reaching behaviors of animals, while the authors of (Maheswaranathan et al., 2019b) used fixed point analyses of RNNs to consider network similarity from a topological point of view.

DYNAMO can be viewed as a form of knowledge distillation (Hinton et al., 2015), since the outputs of the base models serve as targets in the optimization of the meta-model. However, unlike typical instances of knowledge distillation involving an ensemble of teacher networks (Hinton et al., 2015; Fukuda et al., 2017) where individual model predictions are averaged to provide more accurate target labels for the student, our approach instead aims to preserve the dynamics and outputs of each individual base model. FitNets (Romero, 2015) employ a form a knowledge distillation using maps between hidden representations to guide learning; this is similar to our use of hidden state maps in training the meta-model.

Our treatment of the model embedding vectors $\{\theta_n\}$ as learnable parameters is similar to the approach used in Generative Latent Optimization (Bojanowski et al., 2018), which jointly optimizes the image generator network and the latent vectors corresponding each image in the training set. Bojanowski et al. (2018) find that the principal components of the image representation space found by GLO are semantically meaningful. We likewise find that the principal components of the model embeddings found by DYNAMO are discriminative between subsets of models.

Unlike methods such as hypernetworks (Ha et al., 2017) and LEO (Rusu et al., 2019) that use a model to generate parameters for a separate network, our approach does not attempt to reproduce the parameters of the base models in the collection. Instead, the meta-model aims to reproduce only the hidden states and outputs of a base model when conditioned on the corresponding embedding vector.

The core focus of this work also differs from that of the meta-learning literature (Santoro et al., 2016; Ravi & Larochelle, 2017; Finn et al., 2017; Munkhdalai & Yu, 2017), which primarily concerns the problem of few-shot adaptation when presented with data from a new task. Our empirical study centers on the post-hoc analysis of a given collection of models, which may or may not have been trained on different tasks. However, we remark that our exploration of optimization in low-dimensional model embedding space is related to LEO (Rusu et al., 2019), where a compressed model representation is leveraged for efficient meta-learning.

## 4 EMPIRICAL RESULTS

In this section, we describe the results of our empirical study of meta-models trained using DYNAMO on collections of RNNs trained on NLP tasks, and collections of CNNs trained for image classification. For RNN base models, we parameterize the meta-model as a GRU where the model embedding vector $\theta$ is presented as an additional input in each time step. For CNN base models, we use a ResNet meta-model where $\theta$ is an additional input for each ResNet block. In all our experiments, we hold out half the available training data for use as unlabeled data for training the meta-model; the base models were trained on the remaining training data (or a fraction thereof).[1] By default, we set the output loss hyperparameter $\lambda$ to 1. We defer further details on model architectures and training to the Appendix.

### 4.1 VISUALIZING MODEL SIMILARITY IN EMBEDDING SPACE

The base model embeddings $\{\theta_n\}_{n=1}^N$ can be used for cluster analysis to evaluate the similarity structure of a collection of models. We illustrate this use case of DYNAMO via a series of example applications on NLP and vision tasks. Figure 3 shows model embeddings learned by DYNAMO on collections of RNNs. In these plots, we observe a clear separation of these networks according to the size of the available training data, the RNN model architecture used, and the specific NLP task used for training. By computing the eigenvalues of the covariance matrix corresponding to the $N$ model embeddings, we obtain a measure of the intrinsic dimensionality of the corresponding collections of

---

[1] For example, our IMDB sentiment base models trained on 100% of the available training data were trained on 12,500 examples, with the remaining 12,500 examples used as unlabeled data for training the meta-model.

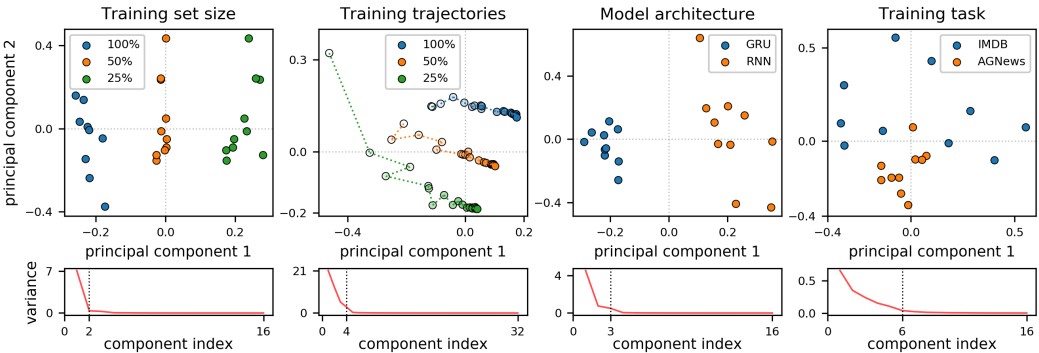

Figure 3: **PCA plots of DYNAMO model embeddings on collections of RNNs.** From left to right: **(1)** The model embedding space for GRUs trained on IMDB sentiment classification dataset (Maas et al., 2011) with varying training set sizes (100%, 50% and 25% of the training data); **(2)** training trajectories of sentiment classification GRUs over 20 epochs, with each point corresponding to an epoch of training (low-opacity points indicate networks early in training); **(3)** two RNN architectures trained for IMDB sentiment classification (GRUs and vanilla RNNs); **(4)** GRUs trained on two NLP tasks: IMDB sentiment classification and AG News classification (Zhang et al., 2015). The second row shows the spectrum for each set of embeddings, with a dotted line indicating the number of components needed to explain 95% of the variance.

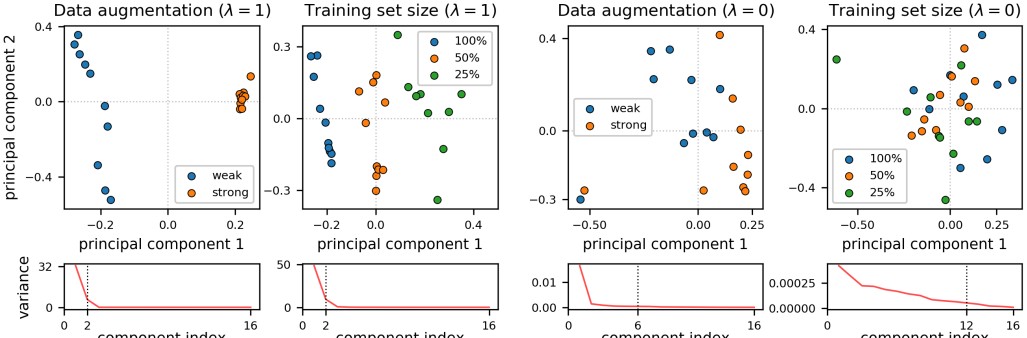

Figure 4: **PCA plots of DYNAMO model embeddings on collections of ResNet-34s trained on CIFAR-100. Left two panels:** Model embeddings cluster according to the size of the training dataset and the data augmentation policy used for training. **Right two panels:** When trained only by comparing hidden representations (i.e., with output loss weight $\lambda = 0$), DYNAMO does not identify a clustering effect when varying the training dataset size. In the case of differing data augmentation policies, there is a weak clustering effect that suggests a consistent difference in feature representation.

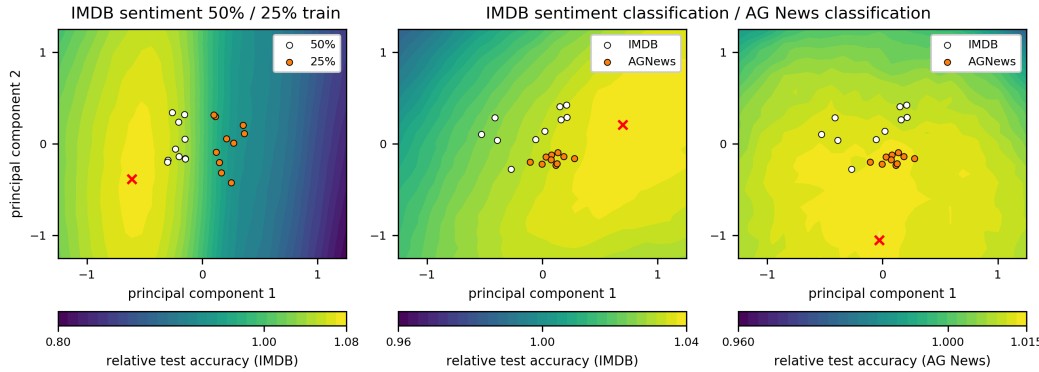

Figure 6: **Meta-model test accuracies over model embedding space.** We plot the relative test accuracies (normalized by the maximal test accuracy of the base models) realized by meta-models for GRUs trained on IMDB sentiment classification with varying training set size **(left)**, and for GRUs trained on IMDB and on AG News classification **(right)**. In these examples, the embedding vectors that maximize test accuracy (marked with an ✗) do not correspond to any single base model, suggesting that meta-models are capable of generalizing beyond the base models used for training.

models. In Figure 3, we find that 2 to 6 components are sufficient to explain $95\%$ of the variance in the model embeddings.

By tuning the output loss hyperparameter $\lambda$ in equation 4, we can adjust the degree of emphasis placed on reproducing the hidden dynamics of the base networks versus their outputs. We demonstrate this effect in Figure 4: with $\lambda = 1$, we observe clear separation of ResNet-34 models trained on CIFAR-100 with different data augmentation policies ("weak", with shifts and horizontal flips vs. "strong", with RandAugment (Cubuk et al., 2020)), and with different training set sizes. In contrast, with $\lambda = 0$, we observe a weak clustering effect with differing data augmentation, and no detectable separation with different training set sizes. We infer that the change in data augmentation policy re-

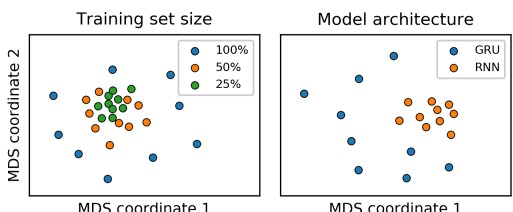

Figure 5: 2D multidimensional scaling (MDS) embeddings of the SVCCA pairwise representational distances between RNN models trained on the IMDB sentiment dataset.

sults in a larger difference in the learned feature representations than the change in training set size. In Appendix B, we illustrate the effect of setting $\lambda = 0$ for RNN models.

We additionally compare the embeddings obtained using DYNAMO to those derived from SVCCA (Raghu et al., 2017), a pairwise comparison technique that aligns the representations produced by a pair of networks using canonical correlation analysis (CCA). In Figure 5, we plot the 2D embeddings obtained using multidimensional scaling (MDS) on the pairwise distance matrix computed using SVCCA (modified to output $L^2$ distances instead of correlations). Unlike the principal components of the DYNAMO model embeddings plotted in Figure 3, the MDS coordinates are not semantically interpretable. Additionally, the cluster structure of the collection of GRUs trained with varying training set sizes is less apparent in this representation.

Lastly, we note that DYNAMO allows for flexibility in defining the metric used to compare the hidden states and outputs of the base models with those of the meta-model. In Appendix B.3, we demonstrate the benefit of using the $L^1$ distance for clustering CNN representations.

## 4.2 EXTRAPOLATION BEYOND BASE MODEL EMBEDDINGS

We study the model embedding space corresponding to a trained meta-model by conditioning it on model embedding vectors $\theta$ other than those assigned to the set of base models. Figure 6 visualizes the landscape of test accuracies for two meta-models: (i) a meta-model for 10 GRUs trained with

50% of the IMDB training data, and 10 GRUs trained with 25% of the data, and (ii) a meta-model for 10 IMDB sentiment GRUs and 10 AG News classification GRUs.

We note two particularly salient properties of these plots. First, the test accuracy varies smoothly when interpolating $\theta$ between pairs of base model embeddings—we would in general *not* observe this property when interpolating the parameters of the base GRUs directly, since they were trained with different random initializations and orderings of the training examples. Second, we observe that the embedding vector that realizes the highest test accuracy lies *outside* the convex hull of the base model embeddings. This is perhaps surprising since typical training and inference protocols involve the use of convex combinations of various objects: for instance, averaging of predictions in model ensembles, and averaging of model parameters during training (e.g., using exponential moving averages, or with Stochastic Weight Averaging (Izmailov et al., 2018)). This extrapolatory phenomenon suggests that DYNAMO is able to derive a low-dimensional manifold of models that generalizes beyond the behavior of the base models used for training the meta-model.

### 4.3    SEMI-SUPERVISED LEARNING IN MODEL EMBEDDING SPACE

The existence of model embeddings that improve on the accuracy of the base models suggests a natural semi-supervised learning (SSL) procedure involving a trained meta-model. In particular, we minimize the loss incurred by the meta-model on a small set of additional labeled examples by optimizing the value of $\theta$. This is done by backpropagating gradients through the meta-model, with the meta-model parameters held fixed. Figure 7 shows the result of this procedure on an IMDB sentiment meta-model (previously depicted in Figure 6) with a set of additional labeled examples (disjoint from the test set) of size equal to 1% of the full training set. This procedure successfully finds a $\theta$ that improves on the test accuracy of the best base model by 6% (86.4% vs. 80.3%).

We observe that this SSL procedure achieves lower accuracy when we train the meta-model using fewer base models. In particular, a meta-model only on the 10 GRUs trained with 50% of the training data yields a test accuracy of 85.4%, and a meta-model on only a single GRU out of these 10 yields 81.6%. This result suggests that a diversity of base models helps improve the accuracy achievable by the meta-model.

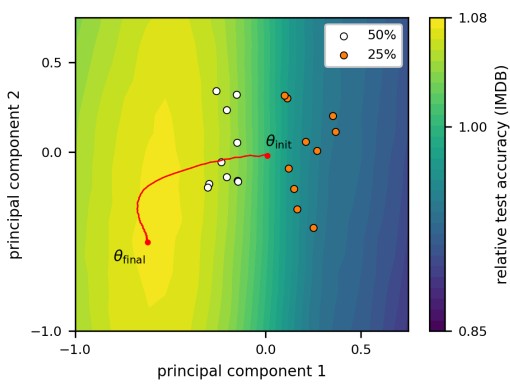

Figure 7: **Semi-supervised learning with low-dimensional model embeddings.** The red line shows the trajectory of 100 SGD iterates in model embedding space, starting from $\theta_{\mathrm{init}} = 0$ and terminating at $\theta_{\mathrm{final}}$. White and orange points indicate the base model embeddings.

## 5    DYNAMICS OF META-MODELS FOR RNNS

In this section we perform an analysis of the dynamical features of the dynamics generated by the meta-models trained on base models that perform the sentiment classification task. The sentiment classification task has a well-understood dynamical structure (Sussillo & Barak, 2013; Maheswaranathan et al., 2019a;b; Aitken et al., 2020) that we can use as a basis for understanding the behavior of a corresponding meta-model. To a first approximation, the sentiment analysis task can be solved by a simple integrator that accumulates sentiment corresponding to each word (positive words such as 'good' or 'fantastic' adding positive sentiment, and negative words such as 'bad' or 'terrible' adding negative sentiment). It has been shown that simple sentiment analysis models approximate this integrator by constructing a line attractor in the space of hidden states. For instance, for the zero input $x^* = \vec{0}$, it has been observed that the dynamical system generated by the map $F_{x^*}(h) = F(x^*, h)$ has a tubular region with very little movement, in the sense that $\|F_{x^*}(h) - h\|_2$ is very small for hidden states $h$ in this region.

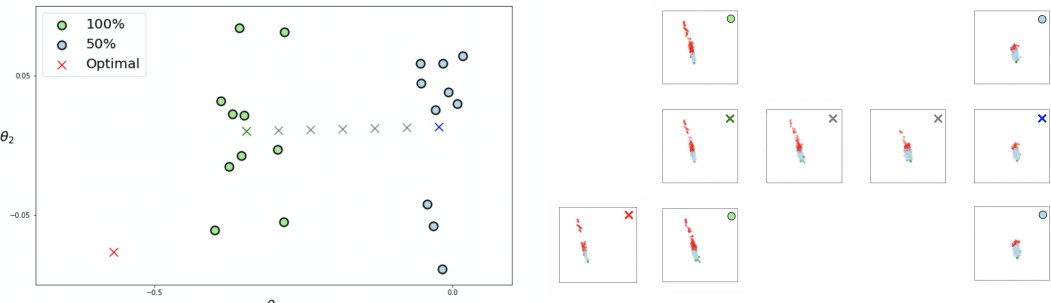

Figure 8: **Model embedding space as a space of line attractors.** We plot the model embeddings of 20 base models trained on the IMDB sentiment classification task **(left)**, along with the centroids of each cluster. The green cluster corresponds to models trained on $100\%$ of the data and the blue cluster corresponds to models trained on a fixed $50\%$ fraction of the training data. A model having better test accuracy than any trained base model is also plotted (marked with an $\times$). For several of the points of interest, we **(right)** find the structure of a line attractor in the hidden state space by computing approximate fixed points of the map $h \mapsto F(x^*, h)$. The line attractors are shown for the point marked with an $\times$, the two centroids, and two interpolated values in the model embedding space. We also chose two values from each cluster to compare the line attractor at the centroid to the models in the structure, corresponding to the top left and bottom right model in each cluster.

To investigate the dynamical behavior of the meta-model space, we trained a meta-model on a set of 20 base models which were themselves trained on the IMDB sentiment analysis task. Of these 20 base models, 10 were trained with $50\%$ of the available training data and the remaining 10 were trained on $100\%$ of the training data. The $\theta$ points corresponding to these base models clustered in the model embedding space according to the amount of training data. In Figure 8 we perform a fixed-point analysis of several models corresponding to points of interest in the model embedding space.

The fixed-point analysis was run according to the procedure described in (Golub & Sussillo, 2018). First we selected a set of candidate hidden states $h_j$ by running the model on a typical batch of inputs. For each hidden state $h_j$ obtained in this way, we used gradient descent on the loss $\|F(x^*, h) - h\|_2^2$ to find the nearest approximate fixed point.

An interesting finding is that the meta-model found line attractor structures that were very geometrically similar for models within a cluster. An interpretation of this result pertaining to *topological conjugacy* in dynamical systems theory is discussed in Section C of the Supplementary Materials. Moreover, we find that the meta-model finds a continuous interpolation between line attractors that are relatively fat and thin (corresponding to models trained on $50\%$ of the data), and models that are elongated (corresponding to models trained on $100\%$ of the data).

## 6 DISCUSSION

We have introduced the algorithm DYNAMO, which maps a set of neural network base models to a low dimensional feature space. Our results show that the model embeddings provided by DYNAMO capture relevant computational features of the base models. Moreover, the model embedding spaces produced by DYNAMO are a sufficiently smooth that model averaging can be performed, and even model extrapolation to reach new models with better performance than any base model. Indeed, in our experiments where the base models were trained on the sentiment analysis task, the model embedding space describes a space of line attractors which vary smoothly in the parameter $\theta$.

We have demonstrated that DYNAMO can be applied broadly to neural networks that have a dynamical structure; for example, we used the demarcation of layers of convolutional neural networks as a proxy for a dynamical time variable. This also suggests possible scientific applications of DYNAMO to dynamical systems arising in nature. A present limitation of DYNAMO is the need for all base models to have the same input structure. For example, one cannot presently utilize DYNAMO

to compare language models trained with different encodings (character-based vs. word-based, for example).

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

## A    META-MODELS FOR CNNS

Our setup in Section 2.2 above can be readily adapted to CNNs, or feedforward neural networks more broadly. For instance, in the case of ResNets, there is a single input $x_0$ followed by a sequence of blocks which operate on different numbers of channels. We let the meta-model likewise be a ResNet with the same block structure. Moreover, we consider the output $b_t$ of the $t$th block in place of the $G_n(h_t)$'s in equation 1, and take the hidden states between the $t$ and $t+1$ blocks to be the $h_t$'s in equation 2. Since each block of a ResNet is distinct, we consider a family of maps $V_{n,t}$ which depend on the base model $n$ and the block layer $t$. Suppose $B$ is the total number of blocks. Then we can more explicitly write

$$\mathcal{L}_{\text{output}}^{\text{CNN}}[\widetilde{F}, \theta] := \frac{1}{B} \sum_{t=1}^{B} d(b_{\theta_n, t}, b_t) \tag{5}$$

$$\mathcal{L}_{\text{hidden}}^{\text{CNN}}[\widetilde{F}, \theta_n, \{V_{n,t}\}_{t=1}^{B}] := \frac{1}{B} \sum_{t=1}^{B} \|V_{n,t}(h_{\theta_n,t}) - h_t\|_2^2 . \tag{6}$$

Then the total loss function accounting for all $N$ of the ResNet base models is

$$\mathcal{L}^{\text{CNN}}[\widetilde{F}, \{\theta_n\}_{n=1}^{N}, \{\{V_{n,t}\}_{t=1}^{B}\}_{n=1}^{N}]$$

$$:= \frac{1}{N} \sum_{n=1}^{N} \mathbb{E}_{x_0 \sim \mathcal{D}_n} \left[ \mathcal{L}_{n,\text{hidden}}^{\text{CNN}}[\widetilde{F}, \theta_n, \{V_{n,t}\}_{t=1}^{B}] + \lambda \, \mathcal{L}_{n,\text{output}}^{\text{CNN}}[\widetilde{F}, \theta_n] \right] \tag{7}$$

where we note that in the expectation value we are only sampling over $x_0$'s in $\mathcal{D}_n$ since $x_0$'s are the only form of input data.

## B    ADDITIONAL EXPERIMENTAL DETAILS

In this section, we provide further details on our experiments as well as additional empirical results.

### B.1    META-MODEL ARCHITECTURES

**RNN architecture.**    We parameterize the meta-model for RNNs as a GRU that takes the model embedding vector as an additional input at each time step. Specifically, the meta-model GRU takes as input the vector $[x_t \,;\, \theta]$ at each time step, where $x_t$ is an input token embedding and $[\,\cdot\,;\,\cdot\,]$ denotes concatenation. The model embedding vector $\theta$ therefore serves as a time-independent bias for the meta-model.

**CNN architecture.**    We parameterize the meta-model for CNNs with a modified ResNet architecture. In each residual block, we use a linear transformation $W$ to map the model embedding vector $\theta$ to the corresponding channel dimension of convolutional layer. We then add the vector $W\theta$ to the channels at each spatial location of the feature map. This design emulates the approach used in our parametrization of the RNN meta-model, with the model embedding $\theta$ serving as a bias term in each residual block. We reuse the weight matrix $W$ for all residual blocks with the same channel dimension.

---

**Algorithm 2** Meta-Model Residual Block
___
  **Input:**  features $x$, model embedding vector $\theta$

  $z = \text{BatchNorm}(\text{Conv}(x))$
  $z = \text{ReLU}(z + W\theta)$
  $z = \text{BatchNorm}(\text{Conv}(x))$
  $z = \text{ReLU}(x + z)$
  **return** $z$

---

The standard ResNet architecture consists of a sequence of four layers, with each layer consisting of a sequence of residual blocks. To compute the hidden state loss $\mathcal{L}_{\text{hidden}}$ in DYNAMO, we compute distances between the output representations of each of these four layers, averaging over the number of channels and spatial locations in each set of features.

## B.2 TRAINING DETAILS

Table 1 lists the hyperparameters used for training our RNN base models and meta-models, and Table 2 lists the hyperparameters used for our CNN base models and meta-models. By default, we use a model embedding dimension of 16. In the case of visualizing the training trajectories of IMDB sentiment GRUs (Figure 3, second column), we use a model embedding dimension of 32 due to the relatively larger number of base models.

| Hyperparameter | Base Model | Meta-Model |
|---|---|---|
| optimizer | AdamW (Loshchilov & Hutter, 2018) | AdamW |
|   - learning rate | $10^{-3}$ ($10^{-4}$ for vanilla RNN) | $10^{-3}$ |
|   - learning rate annealing | cosine with freq. $7/32$ | cosine with freq. $7/32$ |
|   - $\beta$ | $(0.9, 0.999)$ | $(0.9, 0.999)$ |
|   - $\epsilon$ | $10^{-8}$ | $10^{-8}$ |
|   - weight decay | $5 \times 10^{-4}$ | $1 \times 10^{-4}$ |
| number of training epochs | 20 (50 for vanilla RNN) | 100 |
| batch size | 128 | 128 |
| input token embedding dimension | 256 | 256 |
| hidden dimension | 256 | 512 |

Table 1: Hyperparameters used for RNN base models and meta-models.

| Hyperparameter | Base Model | Meta-Model |
|---|---|---|
| optimizer | SGD with Nesterov momentum | SGD with Nesterov momentum |
|   - learning rate | 0.03 | 0.03 |
|   - learning rate annealing | cosine with freq. $7/32$ | cosine with freq. $7/32$ |
|   - momentum | 0.9 | 0.9 |
|   - weight decay | $5 \times 10^{-4}$ | $5 \times 10^{-4}$ |
| number of training batches | $2^{16}$ | $2^{16}$ |
| batch size | 512 | 512 |

Table 2: Hyperparameters used for CNN base models and meta-models.

## B.3 SUPPLEMENTARY EMPIRICAL RESULTS

**Effect of output loss weight.** Figure 9 shows the effect of setting the output loss weight $\lambda = 0$ for meta-models on RNNs. These plots illustrate that model clustering can be performed on the basis of comparing hidden state dynamics alone. We note that the change in the $\lambda$ hyperparameter results in qualitative changes in the resulting clustering. In particular, the model embeddings for GRUs trained on AG News classification (rightmost column of Figure 9) are much more tightly coupled relative to the GRUs trained on the IMDB dataset when $\lambda = 0$. This indicates that the dynamics implemented by the AG News GRUs are much more similar than those implemented by the IMDB sentiment GRUs.

**Clustering with other loss functions.** As noted in Sec. 4.1, the loss functions used to compare hidden states and outputs can be easily changed to better match the characteristics of the base models under consideration. We demonstrate the benefit of this additional flexibility by replacing the $L^2$ distance in $\mathcal{L}_{\text{hidden}}$ (equation 6) with the $L^1$ distance for comparing the intermediate representations of ResNets. This choice is motivated by the observation that the use of the ReLU nonlinearity results in sparse representations, which suggests the use of the $L^1$ metric. Figure 10 shows a comparison between these two distance functions in the case of ResNet-34 models trained on CIFAR-100 with "weak" data augmentation (random shifts and horizontal flips) and with "strong" data augmentation (RandAugment). Here, we parameterized the meta-model with a ResNet-50 architecture. The use of the $L^1$ distance results in a clearer separation between the two sets of models. This is reflected qualitatively in the distribution of the base model embeddings in model embedding space, and quantitatively in the relative scale of the variance captured by the first principal component.

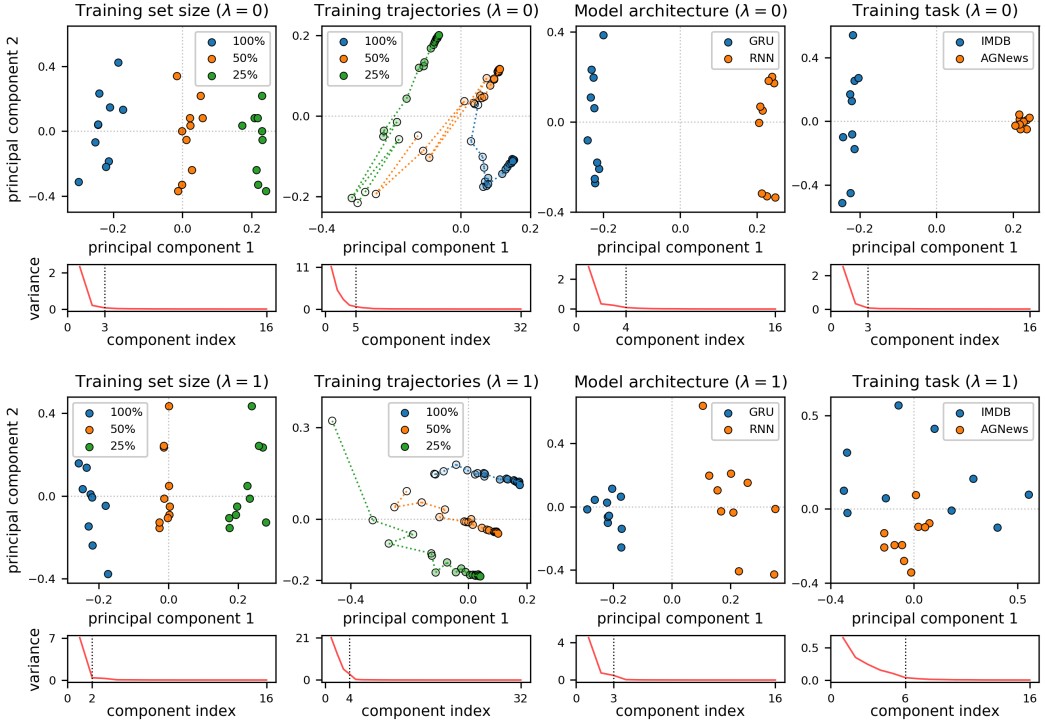

Figure 9: **PCA plots of DYNAMO model embeddings on collections of RNNs with output loss weight $\lambda = 0$.** For ease of comparison, we have also included the plots from Figure 3 with $\lambda = 1$.

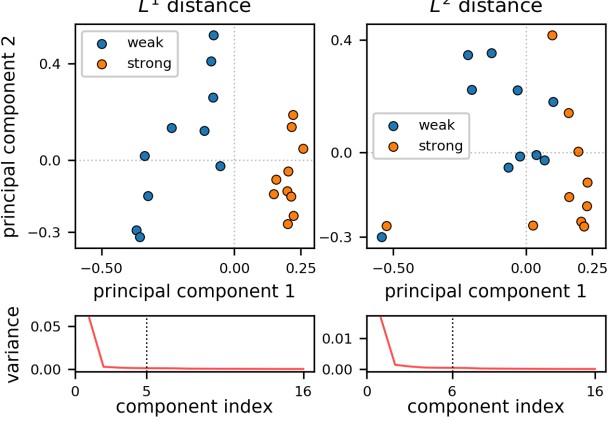

Figure 10: CIFAR-100 ResNet-34 model embeddings using $L^1$ distance to compare intermediate representations **(left)** vs. $L^2$ distance **(right)**. The use of the $L^1$ distance results in a clearer separation between the two sets of models.

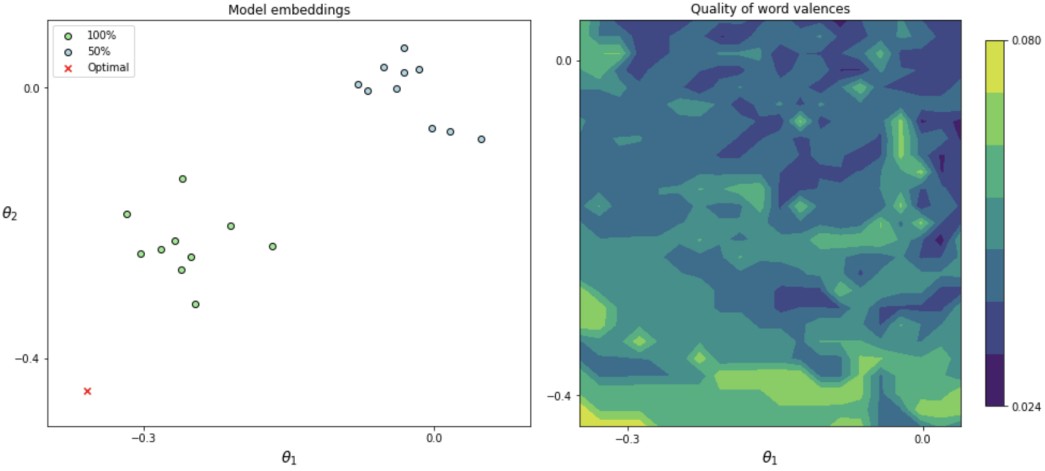

Figure 11: A map of the word scores (described in equation 8) as a function of the parameter in model embedding space. Higher scores indicate models that should have better interpretations of words. There is a noisy but discernible trend that the score increases as $\theta_2$ decreases (and is highest near the value of the optimal model embedding).

### B.4 FURTHER INVESTIGATION OF THE DYNAMICS OF SENTIMENT ANALYSIS

Recall that our trained RNN's implemented sentiment analysis via line attractor dynamics, in which inputted words kicked the hidden state in the 'positive' or 'negative' direction along the line attractor according to the valence of the word (i.e. how positive or negative the word is). Figure 11 investigates how valences assigned to words change as we scan across model embedding space. Given an RNN corresponding to a value $\theta$, we first find a fixed point $h^*$ with neutral readout $G(h^*) \approx 0$. For this fixed point, we compute $G(F(x, h^*))$ for a variety of word inputs $x$. To produce a "score" for the model, we compute

$$\text{Score}(h^*) = \sum_{x \in W_{\text{positive}}} G(F(x, h^*)) - \sum_{x \in W_{\text{negative}}} G(F(x, h^*)) - \sum_{x \in W_{\text{neutral}}} |G(F(x, h^*))|, \quad (8)$$

where the set of positive words $W_{\text{positive}}$, negative words $W_{\text{negative}}$, and neutral words $W_{\text{neutral}}$ are listed in Table 3. Note that here we are not analyzing context effects, for instance how the string 'not terrible' would be rendered into a net-positive valence.

| Positive words | good, awesome, terrific, exciting, fantastic, amazing, fine, superior, outstanding, superb, magnificent, marvelous, exceptional, tremendous |
|---|---|
| Negative words | bad, awful, horrible, terrible, poor, inferior, unacceptable, shoddy, atrocious, crap, rubbish, garbage |
| Neutral words | it, the, is, a, if, then, are, were, can, will, has, had, been, was, when, who, to, what |

Table 3: Lists of good, negative, and neutral words selected to assess the "word valence" quality of a model.

## C  META-MODELS AND TOPOLOGICAL CONJUGACY

In this section we describe the notion of topological conjugacy, its relationship to the loss function $\mathcal{L}_{\text{hidden}}$, and provide speculation as to the interpretation of the results from Section 5.

A topological conjugacy (Katok & Hasselblatt, 1997) between two dynamical systems defined by maps $F : X \to X$ and $E : Y \to Y$ is a homeomorphism $\Phi : X \to Y$ satisfying

$$F(x) = (\Phi^{-1} \circ E \circ \Phi)(x). \quad (9)$$

Note that the dynamical systems $F$ and $E$ can have distinct domains. The significance of the relationship in equation 9 is that dynamics obtained from iterated applications of the map $F$ and $E$ are

related to each other by the formula

$$F^n(x) = (\Phi^{-1} \circ E^n \circ \Phi)(x).$$

Thus, if $F$ and $E$ are topologically conjugate, then their iterates are also topologically conjugate and this means that the dynamics are related by a change of variables. Notice that topological conjugacy is an equivalence relation; as such the transitive property tells us that if $F \sim E$ and $E \sim D$ then $F \sim D$.

The notion of topological conjugacy is an important motivation for defining the loss function $\mathcal{L}_{\text{hidden}}$, which we recall is given by

$$\mathcal{L}_{\text{hidden}}[\mathcal{F}, \theta, V] := \frac{1}{T} \sum_{t=1}^{T} \|V(h_{\theta,t}) - h_t\|_2^2,$$

where the hidden states $h_t$ are dynamics obtained from a base model $F$, the hidden states $h_{\theta_t}$ are dynamics obtained from the meta-model $\widetilde{F}$ with parameter $\theta$, and $V$ is the map from the meta-model hidden states to the base model hidden states. One way of obtaining zero loss is to find a topological conjugacy with map $V$ between the meta-model $\widetilde{F}$ (at fixed $\theta$) and the base model $F$, meaning a relationship of the form

$$F(x, h) = (V \circ \widetilde{F})(\theta, x, V^{-1}h).$$

It is convenient to define the notation $F_x(h) := F(x, h)$ and $\widetilde{F}_{\theta,x}(h) := \widetilde{F}(\theta, x, h)$. Then we have

$$h_t = (F_{x_t} \circ F_{x_{t-1}} \circ \cdots \circ F_{x_1})(h_0) = (V \circ \widetilde{F}_{\theta,x_t} \circ \widetilde{F}_{\theta,x_{t-1}} \circ \cdots \circ \widetilde{F}_{\theta,x_1})(V^{-1}h_0) = V h_{\theta,t}$$

with $h_{\theta,0} = V^{-1}h_0$, so that $\mathcal{L}_{\text{hidden}}[\mathcal{F}, \theta, V] = 0$ would hold. As depicted in Figure 2, to obtain zero loss it would actually suffice to find a weaker relationship of the form

$$F(x, Vh) = (V \circ \widetilde{F})(\theta, x, h).$$

The difference here is that $V$ need not be invertible.

Using the language of topological conjugacy, we can describe a speculative but plausible interpretation of the results of Section 5. In that section we observed that models from the same cluster had very similar dynamical features and performed similarly to the model average of the cluster. This suggests that for each model $F_n$ in the same cluster, we have

$$F_n(x, V_n h) \approx (V_n \circ F)(\bar{\theta}, x, h)$$

where $\bar{\theta}$ is the centroid of the cluster to which $F_n$ belongs. Note that here we replaced $\theta_n$ with $\bar{\theta}$, thus assuming both that $\mathcal{L}_{\text{hidden}}$ is small and that $\theta_n$ is sufficiently close to $\bar{\theta}$. Second, making the hypothesis that there exists an inverse $V_n^{-1}$ to the map $V_n$, the map $V_n$ may provide a topological conjugacy between the base model $F_n$ and the meta-model $\widetilde{F}_{\bar{\theta}}$ evaluated at $\bar{\theta}$. Assuming further that our assumptions hold for all models in the cluster, using the transitivity of topological conjugacy we would conclude that base models belonging to the same cluster are topologically conjugate to one another. This would justify the intuition suggested by Figure 8 that DYNAMO clusters models according to commonalities of topological structures of dynamics.

