# OpenReview forum: "Analyzing Populations of Neural Networks via Dynamical Model Embedding"
_ICLR.cc/2022/Conference — ICLR 2022 Submitted_

### Official Review · Reviewer_dswE · 2021-10-28

**Correctness:** 3
**Technical Novelty And Significance:** 3
**Empirical Novelty And Significance:** 2
**Recommendation:** 5
**Confidence:** 4

**Main Review:**

Quantifying the similarity between two different neural networks trained for the same task is in my opinion an interesting topic which, as correctly mentioned by the authors, has already attracted significant attention. Many methods have been developed to describe the similarity of hidden representations, some of which are cited in the manuscript. In my point of view an open problem in the field is the comparison of the hidden states in  different architectures trained on the same task and dataset.

 My concern is that it is not clear if the approach offers a solution to this problem. The examples presented in fig 3 compare networks trained using the same architecture and different training sets, or different training schedules. In these examples, the metamodel has the same architecture of the networks that it describes. Panel 3 presents an analysis of a  GRU and a vanilla RNN, but in that case the metamodel is just another GRU. This gives the impression that the metamodel should just be an extension of all the models which it should describe, and as expressive as the most expressive network which is considered. The theta parameter practically controls which part of the "ruling network" should be switched off in order to reproduce the behavior of each target network. Is this impression correct?

The key question is if it  is possible to use the approach to make a step forward in the understanding the relationship between hidden representations coded in qualitatively different manner comparing, say  a CNN and a transformer, both trained for image recognition. Or, at least, a network with a VGG and a resnet architecture.

**Summary Of The Paper:**

The manuscript proposes to analyze the similarity of different neural networks trained to solve the same task by developing a meta-model capable of reproducing/emulating both the output and the hidden representation of each individual network.  This meta-model is parametrized by a set of parameters which is common to all the networks, and by other parameters which are network-specific (theta). In this way, each network will correspond to a different theta, and one can analyze the global features of the set of networks by unsupervised manifold learning in the  theta-space.


**Summary Of The Review:**

I think that the approach is smart and potentially interesting, but the empirical evidence used to demonstrate its usefulness is not fully convincing. In all cases defining which  hidden states should be compared  is  easy (or even trivial, if the architecture is really the same), but in these conditions other approaches for comparing the representations can be used as well.

---

> ### Author Response · Authors · 2021-11-19
> **Reply to Reviewer dswE**
>
> We thank the reviewer for their time and valuable comments, and are appreciative for their interest in our approach.  A point-by-point response to the reviewer’s comments is below.
> 1. “Quantifying the similarity between two different neural networks trained for the same task is in my opinion an interesting topic [...] My concern is that it is not clear if the approach offers a solution to this problem“
> - Our method indeed allows us to compare pairs of different networks trained for the same task.  In particular, we can compute pairwise distances between networks represented in the model embedding manifold, thus providing a quantitative measure of similarity.  This notion of similarity is interesting in part because it is not so sensitive to the architectures and weights of the networks being compared, but rather their hidden dynamics and the similarities of their outputs (when given the same input data).
> - We further emphasize that the problem we consider is strictly more general than a pairwise comparison of networks.  We can consider the relative arrangements of k-tuples of networks (pairs correspond to k = 2), and more broadly how a family of networks is geometrically situated in the model embedding manifold.  Moreover, the ability to perform interpolation and extrapolation of the networks in the model embedding space, i.e. the ability to render any point in the model embedding space into an operable neural network, provides us with a more refined notion of comparison of networks.  For instance, we can make sense of and quantify the properties of a neural network halfway between two neural networks, where ‘halfway’ is defined according to the geometry of the model embedding space.
> 2. “This gives the impression that the metamodel should just be an extension of all the models which it should describe, and as expressive as the most expressive network which is considered. The theta parameter practically controls which part of the "ruling network" should be switched off in order to reproduce the behavior of each target network. Is this impression correct?”
> - This may be a possibility but we do not think this is what is happening.  We want the meta-model to synthesize and distill the salient dynamics of the base models (i.e. ‘target networks’), not merely to ‘store’ all of them simultaneously.  Accordingly, we only want the metamodel to be as (or only slightly more) expressive as a single base model, so that the metamodel is forced to find commonalities between the base models in order to minimize the loss of the DYNAMO algorithm.  This is implemented by having the meta-model have a number of parameters which is much smaller than the parameters of all of the base models combined (and is in fact comparable to a single base model).

---

> > ### Comment · Reviewer_dswE · 2021-11-22
> > **possibly incomplete response**
> >
> > I think that the reply does not address a point which I consider very important, which I rewrite here:
> > "The key question is if it is possible to use the approach to make a step forward in the understanding the relationship between hidden representations coded in qualitatively different manner comparing, say a CNN and a transformer, both trained for image recognition. Or, at least, a network with a VGG and a resnet architecture.You did not reply to my key question: ""

---

> > > ### Author Response · Authors · 2021-11-22
> > > **Additional Reply to Reviewer dswE**
> > >
> > > While we have successfully performed clustering experiments with different architectures, for instance a simultaneously clustering of GRU's and vanilla RNN's, we do not compare CNN's and transformers.  (However, we do examine CNN's by themselves.)  It is plausible that such a comparison would work, but it would require a deeper understanding of how to compare the hidden dynamics of qualitatively different architectures.  To be specific, suppose we were interested in comparing a feedforward network with a recurrent network.  Each of them has a notion of hidden state: for the feedforward network we can look at the hidden state at different layers; for the RNN we can look at the hidden state after each iteration.  However, it is not obvious which hidden state of the feedforward network should be compared with which hidden state of the RNN.  That is, there is what we call a 'time alignment' problem for comparing the dynamics.  This time alignment problem is absent when comparing, say, RNN's to other RNN's for language tasks; the reason is that we can compare the hidden states of each RNN after each iteration, and so the iterations provide a type of common clock for the dynamics.  We do not, at present, know how to solve the more general 'time alignment' problem, and this is beyond the scope of the present work.

---

### Official Review · Reviewer_k7ES · 2021-10-31

**Correctness:** 3
**Technical Novelty And Significance:** 2
**Empirical Novelty And Significance:** 2
**Recommendation:** 5
**Confidence:** 4

**Main Review:**

The main strengths of the paper are that the proposed approach is super simple, it depends on a mild set of constraints, and it is quite general. Indeed, the authors show how their idea can be implemented for RNN and CNN models.

The main weaknesses of the paper are the lack of comparison and the lukewarm results. In particular, only Fig. 5 shows a comparison of the proposed approach. However, MDS is computed instead of PCA, so it is not possible to truly evaluate the comparison with respect to Fig. 2.  Moreover, the discussion on related work at the first paragraph of Sec. 3 is somewhat unsatisfying. The main limitation that was mentioned is the O(N^2) computational cost. However, no analysis was provided of the computational resources requires for the proposed method.

Also, the results in Figs. 2 & 3 could potentially have been produced using other approaches, among those is even simple PCA on the network weights/hidden-states/other information (possibly padded to match dimensions). For instance, the third panel in Fig. 3 is somewhat expected, given that the underlying structure of RNN/GRU is significantly from one another.

Another issue is that most results are presented on the sentiment analysis (SA) problem. While SA is a great task for analysis and illustration, it is somewhat a toy example. Thus, it is not clear whether the results presented in this work extend to more challenging tasks solved using RNN models.

Fig. 4 is not entirely clear to me, and I am not sure what can be inferred from it. My impression is that the hidden states of the model are not representative enough, but I might be wrong. Please clarify.

Fig. 6 is interesting and I would like to understand it better. In particular, do you observe a similar phenomenon when using 100% of the data (instead of 25%/50%). Can you improve the state-of-the-art for this model+task using this approach? What is the difference between the middle and right panels? If you sample in 2D, how are the other dominant dimensions are being set? What are the baseline accuracies?

I am not sure what the results in Fig. 8 show. Please elaborate.

minor comments:

	- I could not find a formal definition of the term "manifold". However, this term is used quite heavily in the manuscript, and thus I would expect at least an intuitive explanation for what it means. For instance, the first paragraph in the introduction claims neural networks form a manifold due to their continuously tunable weights. If this is the case, what about the initial network? Is it also part of this manifold of networks? Asked differently, what is not part of this manifold?

	- It is claimed that two points in the meta-model embedding space are nearby if they correspond to neural networks with similar dynamics. This is only intuitively implied. Is there a formal guarantee? If not, I suggest to tone-down this claim, as most of the results are focused on a narrow set of architectures.

	- A similar concern is regarding the convex combination of theta vectors. In particular, even in binary classification problems, the decision boundary is somewhat complex, let alone in a manifold of tasks. It would be interesting to better understand the boundary of the learned manifold.

	- The tasks explored in this work are classification problems. What about regression tasks?

	- The dynamical system studied in App. C is not a proper conjugate system per your definition as it has a different domain.

**Summary Of The Paper:**

This paper introduces a new approach to studying the population of trained neural networks. Specifically, the author's design and train a meta-model which when given a vector, the meta-model emulates the model associated with the given vector. The authors present a few applications of their framework, including clustering and semi-supervised learning.


**Summary Of The Review:**

Overall, this paper presents an intriguing idea of exploring the space of learnt networks. The simplicity and generality of the approach are encouraging. The paper could be significantly improved with an extensive empirical evaluation and comparison, and potentially with formal guarantees.

---

> ### Author Response · Authors · 2021-11-19
> **Reply to Reviewer k7ES (Part 1)**
>
> We thank the reviewer for their time and valuable comments.  We appreciate the reviewer’s recognition of the economy and generality of our approach, and its promise for exploring the space of neural networks in a useful manner.  A point-by-point response to the reviewer’s comments is below.
> 1. “In particular, only Fig. 5 shows a comparison of the proposed approach. However, MDS is computed instead of PCA, so it is not possible to truly evaluate the comparison with respect to Fig. 2.”
> - It seems this comment is based on a misunderstanding of SVCCA, since it does not produce any data which can be used as inputs to the PCA algorithm.  In particular, the SVCCA algorithm only provides a pairwise similarity measure.  As such, the only data provided by the algorithm is a pairwise distance matrix D_ij, where the ij entry is the ‘distance’ between (the representations produced by) the ith network and the jth network in our set.  If there are N networks, D_ij is an N x N matrix.  We then run this through the MDS algorithm to produce points in R^2 which match the pairwise distances in the D_ij matrix as well as possible.
> 2. “Also, the results in Figs. 2 & 3 could potentially have been produced using other approaches, among those is even simple PCA on the network weights/hidden-states/other information (possibly padded to match dimensions).  For instance, the third panel in Fig. 3 is somewhat expected, given that the underlying structure of RNN/GRU is significantly from one another.”
> - To the contrary, PCA on the network weights and hidden states would not yield similar results as in Figs. 3 & 4. To see this, consider two RNNs that are identical up to a permutation of the neurons in their hidden states. Our method allows for both of these networks to be associated with the same theta value, due to the learned per-network transformations V_i. In contrast, PCA does not include any mechanism to resolve this difference between these two functionally identical networks.
> 3. “Another issue is that most results are presented on the sentiment analysis (SA) problem. While SA is a great task for analysis and illustration, it is somewhat a toy example. Thus, it is not clear whether the results presented in this work extend to more challenging tasks solved using RNN models.”
> - As you note, we selected the sentiment analysis task due to its relative simplicity for analysis, and due to prior work that studies the dynamical properties of RNNs trained for this task.
> - We remark that in addition to sentiment analysis, we also consider a text classification on the AG News dataset (Figs 3 & 6), as well as image classification tasks using ResNets.
> 4. “Fig. 4 is not entirely clear to me, and I am not sure what can be inferred from it. My impression is that the hidden states of the model are not representative enough, but I might be wrong. Please clarify.”
> - Fig. 4 serves to illustrate the influence of the output loss weight hyperparameter \lambda on the obtained base model embeddings. Recall that when \lambda = 1, the meta-model is trained to match both the hidden and output states of the base models, and when \lambda = 0, it is trained to match only the hidden states. In Fig. 4, we find that when comparing the embeddings obtained with \lambda = 1 and \lambda = 0, there remains a qualitative separation between the embeddings corresponding to the data augmentation strategy, whereas there is no clear separation in the case of varying training set size. This indicates that the method is able to detect representational differences in the case of differing data augmentation, but not in the setting where the size of the training set is varied.

---

> ### Author Response · Authors · 2021-11-19
> **Reply to Reviewer k7ES (Part 2)**
>
> 5. “Fig. 6 is interesting and I would like to understand it better. In particular, do you observe a similar phenomenon when using 100% of the data (instead of 25%/50%). Can you improve the state-of-the-art for this model+task using this approach? What is the difference between the middle and right panels? If you sample in 2D, how are the other dominant dimensions are being set? What are the baseline accuracies?”
> - With a meta-model trained on 10 base models, each trained on 100% of the training data, we did not observe an improvement in accuracy for theta values other than those corresponding to the base models. Specifically, the best meta-model test accuracy for the theta points corresponding to the base models was 89.4%, and the best test accuracy measured over the grid was also 89.4%. The best test accuracy for the original base models was 85.7% -- thus, there is some improvement which is perhaps attributable to the use of additional unlabeled data for training the meta-model.
> - For comparison, in the case of 25/50%, the best meta-model test accuracy for the theta points corresponding to the base models was 85.2%, compared to the best test accuracy of 86.7% over the theta grid. The best test accuracy for the original base models was 80.3%.
> - In Fig. 6, the middle panel shows the relative test accuracy for the IMDB dataset, and the right panel shows the relative test accuracy for the AG News dataset. For both panels, the meta-model was trained on 10 base models trained on IMDB, and 10 base models trained on AG News.
> - To construct the 2D sampling grid, we use the plane defined the first two principal components of the embedding space.
> 6. “I am not sure what the results in Fig. 8 show. Please elaborate.”
> - Figure 8 shows that the dynamical systems instantiated by the RNN’s in the model embedding manifold all have the structure of (approximate) line attractors.  In other words, the model embedding manifold can be regarded as parameterizing a space of RNN’s which are all (approximate) line attractors.  As we examine values of theta in the space corresponding to line attractors with worse performance on the sentiment analysis task, the associated line attractors degrade (i.e., the fixed points become more diffuse or lumpy).  The Figure demonstrates that our method distills a common dynamical systems representation of all the base models, namely that they all instantiate line attractor dynamics.  As explained in the manuscript, when a new word is inputted into each RNN the hidden state receives a ‘kick’ in either the ‘positive’ or ‘negative’ direction of the line attractor; the point is then drawn back down to the attractor.  The final readout is the dot product along the direction of the line, so that the hidden state being further ‘to the right’ along the line indicates a more positive sentiment whereas the hidden state being further ‘to the left’ along the line indicates a more negative sentiment.
> 7. “I could not find a formal definition of the term "manifold". However, this term is used quite heavily in the manuscript, and thus I would expect at least an intuitive explanation for what it means. For instance, the first paragraph in the introduction claims neural networks form a manifold due to their continuously tunable weights. If this is the case, what about the initial network? Is it also part of this manifold of networks? Asked differently, what is not part of this manifold?”
> - In the revised manuscript we have included an intuitive definition of a manifold for clarity.  To address the comments of the reviewer, suppose we have a network with N weights and biases with a fixed architecture.  Then these parameters describe a point in R^N, to be regarded as a configuration of the given network with its fixed architecture.  But if we consider another network with a different number of weights (or even a different architecture), then the corresponding manifold is distinct.  Said slightly differently, there is no way to continuously change the weights of a fixed network to either (i) change the total number of weights, or (ii) to change the architecture.
> 8. “It is claimed that two points in the meta-model embedding space are nearby if they correspond to neural networks with similar dynamics. This is only intuitively implied. Is there a formal guarantee? If not, I suggest to tone-down this claim, as most of the results are focused on a narrow set of architectures.”
> - The claim made in the manuscript is slightly different, namely that nearby points have similar dynamics.  Thus it is possible that there are non-nearby points which have similar dynamics, although we do not observe this in practice. We have made no formal guarantees.

---

> ### Author Response · Authors · 2021-11-19
> **Reply to Reviewer k7ES (Part 3)**
>
> 9. “A similar concern is regarding the convex combination of theta vectors. In particular, even in binary classification problems, the decision boundary is somewhat complex, let alone in a manifold of tasks. It would be interesting to better understand the boundary of the learned manifold.”
> - Our manifold is not a manifold of tasks, but rather a manifold of neural networks.  As such, the notion of decision boundary does not apply.  When we take a convex combination of theta vectors, we find that the networks corresponding to the ‘averaged’ theta have similar dynamics and performance as the networks corresponding to the thetas being averaged.
> 10. “The tasks explored in this work are classification problems. What about regression tasks?”
> - We have not explored regression tasks in the present manuscript. We agree it is an interesting set of problems to explore.
> 11. “The dynamical system studied in App. C is not a proper conjugate system per your definition as it has a different domain.”
> - Thank you. We have clarified this point in the revised manuscript.  In particular, two dynamical systems with distinct domains can be topologically conjugate.  If we have one dynamical system F : X → X and another E : Y → Y, then a putative topological conjugacy can be implemented by a map which takes X → Y.

---

### Official Review · Reviewer_9dWJ · 2021-11-02

**Correctness:** 2
**Technical Novelty And Significance:** 4
**Empirical Novelty And Significance:** 3
**Recommendation:** 6
**Confidence:** 4

**Main Review:**

The proposed algorithm is novel and interesting. The visualization it provided is meaningful and can be of value to the community. However, I have several concerns about the work at its current state. My main concerns are the lack of proper comparison to the literature, which is explained in detail below.

Pros:
- The proposed model is novel, interesting, and easy to understand.
- The resulting embedding can help better visualize the underlying models.
Cons:
- Requires a large number of trained models to achieve meaningful embedding. Requires additional data.
- I would like a clearer comparison to other similarity measures from the literature.

The paper has several claims on the usefulness of the proposed model. I will go over each of them separately:

Visualization:
Pros: The visualization seems informative and clear, and can be of value to the community.
Cons: Lack clearer comparison to other similarity methods between models. Other methods are only mentioned, but it is unclear which advantages this visualization method has over other ones.

Interpolation:
The ability to interpolate models is new and interesting, and in my opinion is a contribution of this paper, although a bit minor by itself.

Extrapolation:
The authors claim that the meta-model can reach a better performance than any of its underlying models by extrapolation. While this indeed seems like the case, the value of such a finding is unclear to me. To the best of my understanding, the base models trained in Fig 6 saw only a partial amount of the data, while the meta-model saw additional data points during its training. Therefore, it is unsurprising that it could reach enhanced performance.

Semi-supervised learning:
It is hard for me to judge the value of such results, as in my opinion, the comparison here is problematic. As the meta-model sees more data, its accuracy should be compared to other semi-supervised models with similar data, and not the base models.

minor point:
The paper lacks organization: some figures/tables are never referenced in the text (Figure 1 and algorithm 1). The figures sometimes are far from their mention in the text. This made the paper much harder to follow than it should have.


**Summary Of The Paper:**

The authors propose an algorithm that takes several neural networks and outputs an embedding for such networks and a meta-model. This meta-model can take any embedding of a network, and emulate its hidden states and outputs. The authors suggest that these embeddings can measure models similarity, and show how to interpolate and extrapolate between and from the models in the training set.

**Summary Of The Review:**

I find this work novel and interesting. However, I have concerns about the 2 main contributions of the paper, which result in a minor contribution of the paper to the community overall. Therefore, I tend to reject the paper in its current state. However, if I missed something which made these contributions more relevant, please let me know and I'll be more than happy increase my score.

---

> ### Author Response · Authors · 2021-11-19
> **Reply to Reviewer 9dWJ**
>
> We thank the reviewer for their time and valuable comments.  We appreciate that the reviewer found our algorithm to be novel and interesting, including our ability to interpolate between neural networks, and also valued the reviewer’s comments about the value of our visualization methods to the ML community.  A point-by-point response to comments of the reviewer is below.
> 1. “Requires a large number of trained models to achieve meaningful embedding. Requires additional data.”
> - Only a small number of models are required to find a meaningful embedding.  For instance, we considered as few as 3 models.
> 2. “Cons: Lack clearer comparison to other similarity methods between models. Other methods are only mentioned, but it is unclear which advantages this visualization method has over other ones.”
> - We will make this distinction more explicit in the text. In short, existing methods that compare the internal representations of neural networks directly compute measures of similarity or distance between pairs of models. In contrast, our proposed method computes vector representations for each model in the collection. We can compute distances between models using, e.g., the Euclidean distance between these vectors. We note that in addition to this, our method has the following advantages: (1) the low-dimensional “theta” embedding space allows us to reason about the principal axes of variation among base models in the collection, (2) we are able to “execute” new models from this embedding space by conditioning the trained meta-model on new theta vectors, and (3) we gain the additional flexibility of customizing the loss function used for optimizing the meta-model (e.g. using L1 distances when comparing the internal representations of CNNs, as plotted in Fig. 10 in the Appendix).
> 3. “The authors claim that the meta-model can reach a better performance than any of its underlying models by extrapolation. While this indeed seems like the case, the value of such a finding is unclear to me. To the best of my understanding, the base models trained in Fig 6 saw only a partial amount of the data, while the meta-model saw additional data points during its training. Therefore, it is unsurprising that it could reach enhanced performance.”
> - Yes, the base models in Figure 6 were trained using a fraction of the labeled training data available in the corresponding dataset, and the meta-model was trained using held-out, *unlabeled* examples. This training setup is not a fundamental requirement of the method -- the meta-model can also be trained using the same data used to train the base models. In our experiments, we use a hold-out set to train the meta-model since a number of experiments involved base models trained with varying percentages of training examples; in this setting, we did not want to have varying degrees of overlap between the data used to train the meta-model and the data used to train the base models.
> - In Fig. 6, consider the embeddings corresponding to the base models, as denoted by the orange and white points. Indeed, the meta-model, when conditioned on these points, achieves a relatively higher test accuracy than the corresponding base models. This gap may be attributable to the use of the additional held-out data. However, it is surprising that there is a further improvement in test accuracy when one considers embeddings other than those corresponding to the base models. Moreover, these additional gains are realized at points that lie outside the convex hull of the base embeddings, which is perhaps an unexpected phenomenon given the typical practices of model averaging during training and output averaging in model ensembles during inference.
> 4. “Semi-supervised learning: It is hard for me to judge the value of such results, as in my opinion, the comparison here is problematic. As the meta-model sees more data, its accuracy should be compared to other semi-supervised models with similar data, and not the base models.”
> - We agree that further comparisons are needed to fully evaluate the viability of this semi-supervised learning approach. We described this approach to illustrate the fact that the improved theta values shown in Figure 6 can indeed be found in practice using a small amount of additional labeled data, by virtue of the fact that the model embedding space is low-dimensional. This contrasts with how there exist superior solutions (in terms of test accuracy) in the original parameter space of the base models, but these cannot typically be found in practice given a small amount of additional labeled data due to the high dimensionality of the original parameter space.
> 5. “minor point: The paper lacks organization: some figures/tables are never referenced in the text (Figure 1 and algorithm 1). The figures sometimes are far from their mention in the text. This made the paper much harder to follow than it should have.”
> - Thank you for pointing this out.  This is fixed in the revised version of the manuscript.

---

> > ### Comment · Reviewer_9dWJ · 2021-11-28
> > **reply**
> >
> > I agree that it is interesting and surprising that the meta-model can do any meaningful computation outside of the convex hull of the model parameters it was trained on, which indicates some meaningful and interesting embeddings of the trained models.
> > I still think that the comparison to the accuracy of the base models is unfair -- the meta-model did see more examples (although unlabeled ones), hence such accuracies could not be compared in a meaningful way. Drawing conclusions based on these "superior solutions" is problematic.
> >
> > After reading the comments of the other reviewers, I agree with k7ES and dswE about the weaknesses that "it is not clear whether the proposed approach can advance our understanding of different architectures/models which solve the same task", and I think the experiments they offer could greatly benefit any future revision of the paper.
> >
> > I still think this work in its current state is borderline. However, after reading all the comments, I do find the proposed model more interesting and insightful than I initially thought, especially given its low-dimensionality of the embedding space. I'm raising my score to:
> > 6: marginally above the acceptance threshold

---

### Official Review · Reviewer_p41a · 2021-11-03

**Correctness:** 3
**Technical Novelty And Significance:** 2
**Empirical Novelty And Significance:** 2
**Recommendation:** 6
**Confidence:** 4

**Main Review:**

The approach here for ensembling together multiple networks is interesting. The argument for embedding together multiple networks via their theta vectors is weakened a bit by decoders V_i being independently considered in Algorithm 1, rather than have a single decoder V that simply takes theta_i as an input in order to figure out how to decode latent meta-states to individual network hidden states (perhaps with padding, truncation, or some simple projection layer to account for different number of neurons). However, the combined meta-model seems to indeed have value in the ensemble setting, especially given the results in Figures 6-7 and section 4.3, therefore justifying empirically the design used here - but it would be good to see some better reasoning and ablation study to see if and why separate decoders are really needed. To me, this aspect is the most appealing in this work, but it is somewhat understudied, especially since no other ensemble approach is considered here to compare whether training the theta has advantages over more traditional boosting and bagging approaches (applied to the same base models).

For the embedding of networks - the results are rather anecdotal, and while some clear differences are demonstrated, it is hard to see what new insights are provided by them. More fundamentally, the consideration of "manifold of models" is unclear here, even when considered in an intuitive way. This is already seen in the introduction where the authors argue that network weights naturally give rise to some manifold structure. I suppose technically this is true in the same sense that any Euclidean space can be considered as a (rather trivial) manifold, but the same can be said for any parametrized space of functions. In fact, a more accurate description of this point is that neural networks provide a subset of functions that can be parameterized in finite (but very high) dimensional space, rather than be considered in their possibly-infinite dimensional ambient function space, thus enabling their optimization. Similarly, the encoding here maps multiple parametrizations (with potentially varying dimensionalities) into a single parameter space. However, it is unclear in which (non-trivial) sense would this space be considered a manifold. Are similar thetas really providing similar models, functions, or architectures? The results show some very coarse organizations where big differences are more or less exhibited (to somewhat larger degree than SVCCA), but it's unclear if finer differences are really mapped to small changes in theta, or if directions in the theta (PCA) space correspond to some clear or interpretable difference in the base models.

For example, in Figures 3 and 4 (as well as some supplemental figures), several panels are showing the first principal component (capturing much more variance than the rest) aligns with clear differences between configurations (training size, data augmentation, or architecture), but there is also some clear spread in the second component - is this related to some meaningful difference between networks within the same architecture or training regime? Is the varying dimensionality in PCA (based on the 95% explained variance threshold) meaningful? Are individual thetas (prior to PCA projection), or the relations between them via PCA loading, meaningful? The analysis currently provided for these aspects is rather superficial, and it is unclear whether the difference from SVCCA, as demonstrated in Figure 5, is really beneficial or just showing a mere difference between equivalent embeddings. Indeed, while the cluster separation in Figure 5 is perhaps not as clear, or linearly separable, as in figures 3-4, it is quite clear that MDS over SVCCA also organizes the networks according to training size (in concentric rings) or network architecture, and these could probably be nonlinearly separated with an appropriate kernel (and perhaps more than two dimensions).

**Summary Of The Paper:**

This paper proposes an approach to combine together multiple (pretrained) neural networks into a single model that can simulate each of the considered neural networks, both in terms of output and hidden states. The setup here relies on all the considered networks to operate on the same task, and in particular the same input data type (but perhaps with different sampled training sets). The combined meta-model then considers as inputs not only the input data, but also an encoding (which is learned during its training) of each model. This allows model-dependent latent features to be computed by the meta-model, which can then be decoded to hidden states of each network via specialized decoders that can be thought of as auxiliary computational pathways that operate in parallel to the one producing the main output of the meta-model. The model encoding are then used to visualize and explore the relations between different networks (or instantiations), and furthermore, to further tune the combined model towards an interpolation or extrapolation that goes beyond (and seems to outperform) each of the individual networks used to train the meta-model.

**Summary Of The Review:**

The idea in this paper is interesting, and seems promising for an ensemble-learning method interpolating between (or extrapolating from) a collection of neural network base model to a stronger combined meta-model. However, it is not presented or evaluated as such. Instead, this work tries to pose it as a way to learn a "model manifold," and in this context the results are rather superficial and lack clear insights or benefits. Therefore, I would consider it as a borderline paper. It has some potential to be above the acceptance threshold, if it is refocused appropriately and compared to other ensemble methods, but this type of revision might not be sufficiently simple to complete within the tight review schedule for this conference.

---

> ### Author Response · Authors · 2021-11-19
> **Reply to Reviewer p41a (Part 1)**
>
> We thank the reviewer for their time and valuable comments.  We appreciate the reviewer’s interest in our approach, as well as its promise for an ensemble-learning method for interpolation and extrapolation of collections of neural networks.  A point-by-point response to comments of the reviewer is below.
>
> 1. “The argument for embedding together multiple networks via their theta vectors is weakened a bit by decoders V_i being independently considered in Algorithm 1, rather than have a single decoder V that simply takes theta_i as an input”
>
> - A single decoder V would not work. Consider the counterexample where two networks are identical except for a permutation of their neurons. In this case we’d like both to correspond to the sample \theta_i value. A per-network decoder V_i would be able to correct for this permutation.
>
> 2. “However, the combined meta-model seems to indeed have value in the ensemble setting”
>
> - Thanks for the suggestion -- we did not initially approach the idea from the perspective of ensemble methods, but we agree that it is a reasonable direction to consider as well
>
> 3. “the consideration of "manifold of models" is unclear here, even when considered in an intuitive way”
>
> - A manifold is, intuitively, a smooth space of points.  (We have revised the manuscript to provide a more precise definition.)  A family of neural networks specified by having a fixed architecture, say with a total of N weights biases, can be parameterized by a point in R^N.  As such, we can regard an appropriate subset of R^N (matching the ranges of the weights and biases) as comprising a manifold of neural networks with the specified architecture.
>
> - In our work, we consider a different type of parameterization of neural nets in the context of the model embedding manifold.  In this context, we have a neural network (the meta-model) with fixed weights, but the inputs have the form (x,\theta) where \theta is always the same.  The only set of parameters we change are those comprising the \theta vector; thus the space of \theta’s is a manifold which specifies a particular network by partially fixing the inputs to the meta-model.  We can regard the manifold of \theta’s as a manifold of RNN’s, since fixing a \theta in the learned meta-model specifies an RNN.
>
> 4. “Are similar thetas really providing similar models, functions, or architectures? [...]  it's unclear if finer differences are really mapped to small changes in theta, or if directions in the theta (PCA) space correspond to some clear or interpretable difference in the base models [...] there is also some clear spread in the second component - is this related to some meaningful difference between networks within the same architecture or training regime?”
>
> - On ascribing semantic interpretations to directions in PCA space: It’s indeed the case that additional principal components in the theta space capture substantial variance. Since a post-hoc interpretation of these additional directions is difficult, we lack such interpretations for the experiments we ran where we varied one factor in the base models (# training data, architecture, etc.). For experiments where our interventions consisted of two distinct factors (e.g., varying both task and architecture), we found that the first two principal directions respectively corresponded to each of these factors of variation.
>
> - On the amount of spread in the second component: this qualitative feature of the plots should be interpreted with some care. In particular, the visual spread of each component should be considered in combination with the variance captured by that component (as indicated by the lower panels in Figs 3-4). With this caveat in mind, we suggest that the relative spread in the second component between different clusters is indicative of the variability within each sub-group of models. For example, in the 3rd panel from the left of Fig. 3, there is more spread in the vanilla RNN subgroup, and in the leftmost panel of Fig. 4, there is more spread in the subgroup with weaker data augmentation. These relative differences in the spread in the second component align with typical intuitions for each of these settings -- that there is typically more variance in the training outcomes of vanilla RNNs, and that data augmentation engenders a larger effective sample size and therefore further constrains the trained model.
>
> 5. “Is the varying dimensionality in PCA (based on the 95% explained variance threshold) meaningful?”
>
>  - Our intent here is to give an indication of the effective dimensionality of each group of models.

---

> ### Author Response · Authors · 2021-11-19
> **Reply to Reviewer p41a (Part 2)**
>
> 6. “it is unclear whether the difference from SVCCA, as demonstrated in Figure 5, is really beneficial or just showing a mere difference between equivalent embeddings”
>
> - SVCCA yields a pairwise measure of similarity between models. Therefore, in contrast to our method, it does not directly yield an embedding of each model in the collection. While metric embedding methods like MDS can be used to approximate a low-distortion embedding using pairwise distances derived from SVCCA, Fig. 5 demonstrates that the resulting embedding exhibits clear qualitative differences from those obtained from Dynamo. In particular, the separation between sub-groups is less evident in the case of SVCCA+MDS, and while it may be possible to obtain a better separation using a kernel method (as the reviewer suggests), it is unclear how this can be achieved in an unsupervised manner, without the sub-group labels for each base model being made available.
>
> - We emphasize that unlike SVCCA, our method optimizes embeddings for each base model in the collection directly. This helps simplify the model analysis pipeline by obviating the need for additional post-processing steps, such as MDS, for obtaining a final embedding.

---

### Decision · Program_Chairs · 2022-01-20

**Decision:**

Reject

**Comment:**

This paper introduces an algorithm for making a meta-model from an ensemble of models by learning model embedding.

All reviewers appreciated the originality and potential usefulness of the paper. However, they also all think that the work is not completely ready for publication. Both the presentation and quality of the results can be improved.

There are a lot of good feedback in the discussion that can be used to make an important updated version for the next conference.